# Assessing the Influence of Green Innovation on the Market Performance of Small- and Medium-Sized Enterprises

**Shiyong Zheng** [1,2,3], **Xinsen Ye** [1], **Weili Guan** [2,*], **Yuping Yang** [1], **Jiaying Li** [1] and **Biqing Li** [1]

1    School of Business, Guilin University of Electronic Technology, Guilin 541004, China
2    College of Digital Economics, Nanning University, Nanning 530021, China
3    School of Management, Hainan University, Haikou 570228, China
*    Correspondence: author: wlguan@126.com

**Abstract:** Green innovation is a significant component of high-value growth closely linked to China's 14th five-year plan. This research investigates the influence of green innovation on the market performance of small- and medium-sized enterprises (SMEs). The results are based on the primary data gathered via an online questionnaire survey from 453 respondents working for SMEs in China. The structural equation modeling approach is used for data analysis purposes. The research findings highlight that green innovation positively impacts marketing and products. In turn, marketing innovation positively influences product innovation and market performance, and product innovation also significantly boosts market performance. The study's findings lead us to suggest that organizations in developing countries should focus on SMEs' green innovation, which will support them in achieving an effective market performance. The study's limitations are noted so the findings can be interpreted with caution, and directions for future research are outlined for all stakeholders.

**Keywords:** green innovation; market performance; green innovation culture; marketing innovation; product innovation; small- and medium-sized enterprises

## 1. Introduction

Innovation aids in adapting to change and discovering new possibilities; this can provide businesses with a competitive advantage by helping them generate better goods and services for their customers than the competitors. Today, the need to innovate significantly impacts how businesses function [1]. Studies examining the relationship between innovation and firm size are generally interested in the consequences of enhanced market performance (MRP) and new market innovation [2]. Small- and medium-sized enterprises (SMEs) have a stronger capacity to innovate than larger corporations [3]. SMEs play an important part in today's economy due to their size and nature of employment, and beyond that, their capacity to invent new goods and processes significantly impacts nations' economies in an increasingly competitive global market [4]. In that context, improving SMEs' creative talents and knowledge may unlock considerable potential. Taking such approaches to fuel innovation is becoming vital for competitiveness and delivers added benefits for the efficiency and success of the private sector [5].

Even though SMEs appear to be a hot topic today, few studies have investigated their innovativeness in developing economies [5]. When considering this notion in the context of China, it is clear that the importance of SMEs is growing in the country's present market economy [6]. Most SMEs in China were created during the last 30 years, and they have been important to the country's economic success in the decades since it opened to the market economy in the 1980s as part of Deng Xiaoping's market-oriented reforms [7]. Since the economic trends that followed the reforms negatively impacted large state-owned enterprises (SOEs) until the end of 2004, in the late 20th century and the early years following the millennium, many SOEs in China were quickly converted to private SMEs.

Meanwhile, the non-SOE marketing strategy implemented in China aided the establishment and expansion of additional SMEs. Since then, urban collective enterprises (UCEs), town and village enterprises (TVEs), and the private and self-employment sectors have gained popularity in China.

The rise of SMEs has become ever more important for China's economic development. These SMEs not only play a substantial part in the country's fiscal progress, accounting for about 99% of all firms but also greatly contribute to the increased business activity and employment in China [8]. Despite this, SMEs face several obstacles in their quest to innovate effectively. To improve SMEs' ENS, it is vital to investigate the elements currently limiting their innovation [9].

In that context, this study explores the link between Chinese innovation and MRP in China. The research also analyzes the resource-based perspective (RBV) put forward in Terziovski's work. Unlike prior research, this places a certain focus on MRP as a factor affecting a company's success [10]. Furthermore, this study provides clear knowledge of innovation paradigms and how SMEs generate ENS, particularly in China. The findings of this study emphasize the relevance of marketing innovation (MRK) and green innovation culture (GIC) among SMEs for their product innovation (PRD). It is important to remember that innovation is necessary at all phases of competition and that it generates riches in the commercial sector [11]. Since, according to various studies, small businesses spend more money on product development than process development, this research is entirely concerned with the impact of PRD on MRP [12].

To grow by generating new goods and services, businesses must imbue their operations with an innovative culture. If they are to succeed, empirical research shows that such a culture must be established, maintained, and encouraged. Creativity, empowerment, and transformation of the company culture are all catalysts for innovation. To date, the literature focuses on MRK and GIC, and the importance of an innovative culture and the influence of MRK on PRD have not been properly investigated in previous studies [13].

Against that background, this study determines the importance of MRK methods and PRD for boosting ENS. The basic assumption of this study is that MRK is required to improve MRP [14]. The primary contributors to MRP are marketing and PRD strategies. Competition has become a necessary component of market survival, and in that context, innovative activities can help a firm to develop distinctive products and services. These may raise a firm's value, provide advantages, and help the firm stand out from its competitors in such ways that PRD may boost the firm's performance.

This study's focus is thus on the influence of creative activities, such as innovation, on ENS. This research contributes to the current body of knowledge by revealing how SMEs may succeed at PRD by cultivating a distinct innovative culture and MRK [15]. Cases are offered as examples of where SMEs have outperformed large corporations in the marketplace owing to their creation of innovative marketing tactics and goods [16]. The study takes a theoretical approach whereby an RBV is thought of as essential to explain and resolve a model's future issues. In that way, the current study addresses the following research questions: (i) how do market and PRD activities affect SMEs' ENS? (ii) What impact does SMEs' entrepreneurial culture have on their marketing and product development? (iii) How do marketing and PRD combine to impact consumer behavior?

The remainder of the paper is structured as follows. A modeling framework and hypotheses are proposed in the second section. Next, a discussion of the methodology, sample, and measurements is provided in Section 3. Section 4 then offers the results and a discussion. Finally, the last section concludes the study and emphasizes important policy recommendations and directions for further studies.

## 2. Modeling Framework and Study Hypotheses

As previously mentioned, innovation refers to the creation of a new method of doing anything, whether the business is physical (e.g., the manufacture of a new product) or abstract (e.g., the development of new technology) (e.g., the creation of a new theoretical

approach) [17]. It is important in the discovery and development of viable systems of production and living since alternatives to the current way of doing things are required in both situations [18]. Because innovation plays a central role in business development and progress, considerable research has been undertaken that focuses on PRD, its role in SMEs, and market innovation [19].

According to some academics, SMEs' success is influenced by several crucial aspects. According to cross-national research, having a successful marketing strategy, good client connections, and competent management are all aspects that contribute to the success of SMEs [20].

The literature [21] identified seven significant aspects that influence SME's success, including having the capacity to develop and sustain a technological edge; the capacity to recognize and concentrate on one or more market segments; strong top-level leadership; a major "people bonding" mechanism; a capable management team; consumers' strategic alliance; and the strategic use of information technology [21]. Strong leadership, a strong customer and client connection, a supportive and strong management system, the right strategy, market focus, building and maintaining capabilities, and a good customer and client relationship are six key factors that determine a firm's performance, according to a study of Singapore's top 50 SMEs [22]. Another piece of research on PRD yielded insights on supporting actions for various SMEs. They presented evidence in support of three product innovativeness (PI) categories: direct mimics (low PI), product innovators (high PI), and idea innovators (high PI) (medium PI). It also demonstrates that these groups' performance potential varied at the product level but not at the company level [23]. Even though most SMEs can adapt quickly to changing environments and satisfy changing customer wants, many struggle with successful innovation and are more likely than their larger counterparts to face resource and capability constraints [24].

As a result, studies often highlight the elements that negatively influence or obstruct SMEs' market success. Tolerance for ambiguity, risk-taking, environmental strategy and scanning, heterogeneity in the organization, and professionalism in all work areas are some of the primary elements that separate creative from non-innovative SMEs in Korea, according to research conducted in Korea [23]. It is suggested that a lack of funding and low return rates are the major hurdles. It has also been discovered that companies that prioritize research and development (R&D), staff training, and employee networking have a better likelihood of innovating [25]. This has resulted in a higher number of new items, technical advancements in both processes and products, and a stronger emphasis on prototype development. All of these factors appear to indicate that the most significant impediments to SMEs' innovation are a lack of financial resources, low return rates, high cost and risk of innovation, a lack of sufficient information, and a lack of proper training [26]. Although the importance of these criteria varies based on the type of innovation and the context, these are the ones that are frequently mentioned in the research.

China is perhaps one of the best countries where to study innovation and its associated phenomena in detail due to it being the largest evolving economy in the world. The country classifies SMEs mostly based on the number of employees they have, which is often less than 500 in most cases [27]. The definition of a small business in China is complicated because it is based on several factors such as industry category, the number of employees, annual revenue, and total assets, all of which are based on the SMEs, which establishes the guidelines for classifying SMEs.

Based on the above discussion we will analyze the innovation constructs that the research proposes. This study examines the relationships between MRK, culture, PRD, and MRP in Chinese small businesses. As a result, the reason for examining SMEs' creative activities in China is supported by the fact that many SMEs from emerging countries reveal that green innovation and MRP are crucial in boosting SMEs' success. The study analyses innovation research in SMEs to identify the relevant building blocks for enhancing a theoretical model. A resource-based perspective (RBV) describes how internal resources in SMEs influence performance and foster competitiveness. The RBV is a viewpoint

that looks into how superior-performing businesses assign their traits to their employees. Furthermore, the RBV can aid in gaining a deeper understanding of these enterprises' success than other businesses.

The majority of the researchers opined that innovation-related frameworks impact the MRP of SMEs. From an innovative aspect, this method was broadened to embrace the imaginative nature of SMEs. Several studies employed several innovation-related frameworks as prospective variables for confinement in the model in this respect. The company model, competition, culture, and technology are all examples [28]. On the other hand, marketing and innovation are critical to the success of many businesses and have been recognized in several marketing and management publications. As a result, the criteria used in this study include MRK, PRD, and GIC. The study framework is constructed by categorizing innovation-related elements into three major categories and analyzing how these factors influence the MRP of SMEs.

### 2.1. Market Performance

SMEs have revenues, assets, and employees that fall below a certain threshold. The criteria for defining SMEs differ by country and, in some cases, by industry. When it comes to China, the definition of SMEs is quite complex since there seems to be no set ideal. There are specific guidelines about the entire industrial sector enterprises' total assets, including mining, electricity, water, gas, supply, and construction. However, in the case of industries such as retail businesses, restaurants, hotels, and transportation, there seem to be no asset requirements. On the other hand, the recommendations for the industrial sector in the case of SMEs talk about having a maximum of 2000 workers and annual revenue of not more than RMB 300 million. Their total assets should be no more than RMB 400 million.

There are differences between how SMEs operate compared to how large enterprises function that have been extensively discussed in the literature. These differences generally arise when it comes to ownership, resource availability and limitations, decision-making, and the overall size of the organization, as discussed in the paragraph above. This is also the case in China, and so is the very high percentage of failure of SMEs. As in such organizations worldwide, lack of leadership, capital constraints, and resistance to change often lead to SMEs failing too often [29]. Due to this and other market competitive aspects, SMEs are forced to focus on marketing, innovation, and increased productivity. Although SMEs in China have achieved fast and rapid growth in the last decade and have also contributed extensively to the development of the country, weak linkages, weak technological innovation, the external market, and limited financing, most of these aspects as having been discussed above, have limited the growth of SMEs [30]. This emphasizes the importance of more successful market innovation when trying to sell new items to local and worldwide markets. Here MRP refers to the relationship between market share, sales determinants, product revenue premiums, and service revenue premiums. Previous research has shown a favorable association between innovation and performance. While much of this research focused on the impact of innovation on SME performance, it looked at the impact of innovation on SME MRP very little. Consequently, this study looks at the correlation between various kinds of innovation and SMEs' market success.

### 2.2. Green Innovation Culture

It is necessary to understand the relationship between innovation and culture because the two are very intertwined and dependent on one another. Innovation, in very simple words, refers to something new or a change that is made to an existing product, field, or idea [31]. It is a process through which a domain, product, or item is renewed and brought up to date by using certain techniques, applying new processes, or incorporating new ideas and concepts. However, the introduction of an innovative product or any innovation and its application is not easy unless there exists a culture that supports it. Green innovation in companies generally takes place if companies follow certain practices such as discovering new markets instead of competing rigidly with a few competitors,

diversifying both the workplace and the product, taking input from several sources, i.e., focusing on creativity from a crowd instead of just internal solutions, creating a free space where employees feel comfortable enough to bring up ideas and lastly motivating and appreciating innovation [31].

The innovation culture in China has led to it becoming a hub for SMEs in the world, in addition to becoming the largest emerging economy. China's innovation may be divided into two categories. The government is largely responsible for technical innovation through its support of enterprises through supportive and facilitative policies. This covers projects such as deep-seated sea research, quantum computing advancements, and many others. Another thread is technology-enabled commercial innovation [32]. Factors such as low resistance to change and high entrepreneurial activities also seem very high in probability in the country, further contributing to its innovation culture. As a result, SMEs in China have identified new tactics for establishing new channels and adopting new ways of selling a product that customers appreciate [33]. Along these lines, [34] analyzed the linkage between internal control, environmental investment, and green innovation in the Chinese context. The authors reported a positive linkage between internal control and corporate green innovation based on study findings. In another study, [35] scrutinized the influence of green technology innovation on the financial performance of Chinese listed semiconductor concept stocks. The analysis revealed that proactive green process innovation significantly influences corporate financial performance both in the short-term and long-term. However, proactive green product innovation significantly influences corporate financial performance only in the long term. SMEs can achieve a competitive advantage when it comes to enhancing productivity and marketing methods, as well as obtaining desired outcomes, as a result of their improved innovation culture. As a result, the literature establishes a strong correlation between innovation culture and marketing.

**Hypothesis 1 (H1):** *GIC positively influences the MRK of SMEs.*

Scholars also highlight the influence that GIC has on PRD in SMEs. Hence, to discuss this particular relationship:

**Hypothesis 2 (H2):** *GIC positively influences the PRD of SMEs.*

### 2.3. Marketing Innovation

Marketing refers to a process through which customers are informed about the products or services an organization offers them. It brings tremendous value to an organization's sales and innovation success [36]. Most of the innovative procedures that lead to the success of new products in the market are covered by MRK. Thus, innovation and marketing must at all times go hand in hand. While innovation helps develop products by anticipating the buyer's needs and how they may evolve or change in the future, in MRK, customers' value judgments must be evaluated, and possibilities for unmet consumer demands must be identified so that firms may deliver new inventive goods [37].

PRD is important in marketing because it not only attracts new customers through the promise of something new and better but also enlarges product categories and segments. Thus, a positive relationship is generally seen between MRK and PRD. SMEs in China are known for innovation in their marketing tactics because of the fierce domestic and international competition. Due to the wide range and variety of products available, companies must constantly develop new and better marketing tactics to make their product stand out and be known in the market. This not only empowers a product, even if it is cheap and of low quality, but also increases its reach. We can hypothesize the following based on the strong association between marketing and PRD:

**Hypothesis 3 (H3):** *MRK positively influences the PRD of SMEs.*

Similarly, establishing a solid link between MRK and MRP is simple since innovation in marketing leads to the attraction of new customers while still holding the interest of the previous ones, thus positively impacting ENS [38]. Hence it can be hypothesized that:

**Hypothesis 4 (H4):** *MRK positively influences the MRP of SMEs.*

*2.4. Product Innovation*

While innovation can be regarded in many forms, the focus will be on PRD for this particular part of the paper. PRD is generally associated with introducing a newer and better version of a product in a market, keeping in mind the present and upcoming needs of the consumer base. PRDs are ideally suited for new businesses entering the market since they help gain market share, enhance profitability, and favorably influence the organization's ENS [39].

In another study, [40] examined the relationship between green technology innovation and corporate financial performance using bibliometric analysis. The findings indicated that green innovation and product innovation are considered research hotspots, and China performs best in these fields, followed by Spain and the UK. China is well known for its PRD ventures, most prominently due to the Chinese government's continued stress on evolving technology in products, including its research and development (R&D) spending, which was around 42$ of the US level in 2014 [41]. SMEs in China adopt various methods and techniques to ensure and later implement PRD, including cost, process, and technological innovation. When seen in a larger context, China has created a very small number of novel product inventions [42]. However, Chinese enterprises are shifting from incremental to radical breakthroughs based on their tremendous experience with incremental advances. Huawei is one such company. With its distributed base stations and SingleRAN-based LTE solution for mobile operators, it is beginning to make significant changes.

Another example is the Sanyi Heavy Industry Company, which manufactures the world's most powerful crawler crane [41]. Huawei's exceptional MRP is an example of how PRD helps an SME. To further study this relation, we have come up with another hypothesis which is as follows:

**Hypothesis 5 (H5):** *PRD positively influences the MRP of SMEs.*

Figure 1 represents the research framework of the study.

**Figure 1.** Research framework.

## 3. Materials and Methods

### 3.1. Data Collection

We collected the primary data to study the hypotheses discussed above. For this purpose, we conducted an online questionnaire survey. The CHERRIES checklist was followed for this study during the online questionnaire survey. The Institutional review board of Guilin University of Electronic Technology (protocol code 893-2) approved the study. A random sampling method was employed to select the respondents. The research objective was explained to all of the respondents who participated in the survey. The anonymity of the respondents was assured. The participants provided their written informed consent to participate in this study. Before the actual survey, a pilot survey on a small sample size was performed. A closed survey technique was adopted during the survey. The survey was advertised through emails, and the respondents participated voluntarily during March and April 2022. A total of five hundred sixty-seven questionnaires were sent via email to respondents in different SMEs with a cover letter stating the brief introduction and purpose of the study. We divide SMEs into five categories: technical, hybrid, instructional, functional, and sentinel SMEs. More specifically, these include administrators, proprietors, marketers, R&D managers, original equipment manufacturers, and medical professionals. Individuals from these SMEs were selected through China Stock Market & Accounting Research Database (https://www.gtarsc.com/ accessed on 23 July 2022). As a result, 453 respondents completed the survey in its entirety, with a response rate of 79.8 percent.

There are numerous reasons to concentrate on SMEs in particular. This is because they play an important role in global economic development and wealth accumulation. Moreover, small enterprises promote job creation, resulting in the most dynamic environment in growing markets. Overall, creative activities provide SMEs with the skills they need to reduce product life cycles, boost survival chances, compete, and move forward in a competitive environment. This is true, particularly in upcoming economies, where small enterprises have little resources and find innovation to be an expensive task.

### 3.2. Sample and Selection of Respondents

Table 1 reports sample characteristics. For the entire sample, business owners made up 36.4 percent of the sample, while 26.4 percent of the sample had less than five years of experience. Small businesses account for around 74.6% of the organizations surveyed, with 1 to 50 employees (in terms of the number of people that work there).

**Table 1.** Sample characteristics.

| Characteristics | Frequency |
| :---: | :---: |
| **Position within the company** | |
| Business owner | 36.4 |
| Other | 21.3 |
| Board of directors | 24 |
| HRM Manager | 10.6 |
| Marketing Manager | 5.2 |
| R&D Manager | 2.5 |
| **Work Experience** | |
| Less than five years | 26.4 |
| 6–10 years | 23.6 |
| 11–15 years | 24.2 |
| 6–20 years | 10.6 |
| 21–25 years | 6.8 |
| More than 25 years | 8.4 |
| **Organization's size** | |
| Less than ten employees | 45 |
| 10–30 employees | 14.5 |
| 31–50 employees | 15.1 |
| 51–100 employees | 8.2 |
| 100–300 employees | 9.4 |
| More than 300 employees | 7.8 |

## 4. Results

### 4.1. Scale Validity and Reliability

Structural Equation Modeling (SEM) is a data analytics procedure used to test the hypotheses in the research conducted. The loadings, Cronbach alpha, and composite reliabilities are mentioned in Table 2 below. For all variables in the model, composite reliability was explored. Regarding measure dependability, composite reliability ratings over 0.60 are considered satisfactory. All composite reliabilities were higher than the recommended value of 0.60 (0.831, 0.923, 0.888, and 0.806, respectively).

**Table 2.** Factor loading and discriminant validity.

| Variable | Items | Standard Loadings | Cronbach-$\alpha$ | CR |
|---|---|---|---|---|
| **Green innovation culture** | | | 0.913 | 0.831 |
| | GIC 1 | 0.782 | | |
| | GIC 2 | 0.822 | | |
| | GIC 3 | 0.877 | | |
| | GIC 4 | 0.861 | | |
| | GIC 5 | 0.853 | | |
| **Product innovation** | | | 0.849 | 0.923 |
| | PRD 1 | 0.750 | | |
| | PRD 2 | 0.816 | | |
| | PRD 3 | 0.796 | | |
| | PRD 4 | 0.837 | | |
| | | 0.713 | | |
| **Marketing innovation** | | | 0.915 | 0.888 |
| | MRK 1 | 0.754 | | |
| | MRK 2 | 0.724 | | |
| | MRK 3 | 0.741 | | |
| | MRK 4 | 0.752 | | |
| **Market performance** | | | 0.927 | 0.806 |
| | MRP 1 | 0.723 | | |
| | MRP 2 | 0.743 | | |
| | MRP 3 | 0.686 | | |
| | MRP 4 | 0.704 | | |

**Notes:** CMIN/DF: 2.247; GFI: 0.956; CFI: 0.961; NFI: 0.952; IFI: 0.967; RMSEA: 0.048; AGFI: 0.971; RMR: 0.46.

### 4.2. Measures and Measurement Model Testing

Innovation culture, PRD, MRK, and market success in SMEs are the four elements investigated in this study. The majority of the variables were taken from previous research. The literature of [43,44] provided the MRK variables. The items related to innovation culture were evaluated using a five-item scale based on The Effect of Firm Compensation Structures on the Mobility and Entrepreneurship of Extreme Performers [45,46]. A five-point scale adapted from [47,48] was used to assess PRD. [43] provided the MRP items, captured their measurement items on a five-point scale ranging from 1 = strongly disagree to 5 = strongly agree.

The resulting entries were subjected to confirmatory factor analysis in AMOS 26 using SEM. The goodness of fit indices indicate that the samples were well-fit (CMIN/DF: 2.247; GFI: 0.956; CFI: 0.961; NFI: 0.952; IFI: 0.967; RMSEA: 0.048; AGFI: 0.971; RMR: 0.46).

Correlation analysis was employed to determine the link between the variables. The convergence validity was determined using average variance extracted (AVE) values and item loadings. All of the AVE values for the constructs were more than 0.50, indicating that the latent factors accounted for at least 50 percent of the variation. Using the square root of AVE, the discriminant validity was measured. The outcomes exhibit discriminant validity since the square root of the AVE is larger than its association with other variables [49]. The maximum shared variance (MSV) values are less than the AVE values for all of the variables, further proving the discriminant validity of the model (see Table 3).

**Table 3.** Descriptive statistics and correlations.

| Variable | GIC | PRD | MRK | ENS | AVE | MSV |
|---|---|---|---|---|---|---|
| **GIC** | 0.743 | | | | 0.552 | 0.529 |
| **PRD** | 0.230 | 0.840 | | | 0.705 | 0.122 |
| **MRK** | 0.727 | 0.218 | 0.784 | | 0.614 | 0.529 |
| **ENS** | 0.297 | 0.349 | 0.175 | 0.714 | 0.510 | 0.122 |

### 4.3. Effect Size ($f^2$)

In Table 4, our path model is constructed by excluding a dominant construct every time to identify the effects of an independent variable on the dependent variable. It is proposed that [50], $f^2$ values of 0.02, 0.15, and 0.35 are considered small, medium, and high, respectively. Table 4 shows that GIC has small impact on both MRK ($f^2 = 0.117$) and PRD ($f^2 = 0.016$), while MRK has a medium effect on PRD ($f^2 = 0.157$), and ENS ($f^2 = 0.644$). Lastly, PRD has a large effect on ENS ($f^2 = 0.229$).

**Table 4.** Effect size ($f^2$) statistics for the general model.

| Hypotheses | Hypotheses Paths | $f^2$ | Effect Size |
|---|---|---|---|
| **H1** | GIC → MRK | 0.117 | Small |
| **H2** | GIC → PRD | 0.016 | Small |
| **H3** | MRK → PRD | 0.157 | Medium |
| **H4** | MRK → ENS | 0.644 | Medium |
| **H5** | PRD → ENS | 0.229 | High |

The SEM flowchart is depicted in Figure 2. An optimistic and important association was established ($\beta = 0.043$, $p < 0.05$) between green innovation culture and marketing innovation GIC → MRK. Consequently, we accepted H1. The standardized estimates of GIC with PRD ($\beta = 0.092$, $p < 0.1$) presents a significant linkage between green innovation culture and product innovation. Thus, we accepted H2. Similarly, marketing innovation positively affects product innovation, as the MRK standardized estimates ($\beta = 0.086$, $p < 0.1$) show a significant linkage with PRD. Consequently, we accepted H3. Marketing innovation also positively affects market performance, as the MRK standardized estimates ($\beta = 0.024$, $p < 0.01$) exhibit a significant linkage with ENS. Therefore, H4 was validated. H5 was also accepted as product innovation significantly affect market performance ($\beta = 0.064$, $p < 0.05$). The hypothesis outcomes are reported in Table 5.

**Table 5.** The structural model.

| Hypotheses | Hypotheses Paths | $\beta$-Value | $f$-Value | Result |
|---|---|---|---|---|
| **H1** | GIC → MRK | 0.043 ** | 159.1 *** | Accepted |
| **H2** | GIC → PRD | 0.092 * | 198.8 *** | Accepted |
| **H3** | MRK → PRD | 0.086 * | 253.7 *** | Accepted |
| **H4** | MRK → ENS | 0.024 *** | 187.7 *** | Accepted |
| **H5** | PRD → ENS | 0.064 ** | 352.7 *** | Accepted |

**Notes:** *** $p < 0.01$, ** $p < 0.05$, * $p < 0.1$.

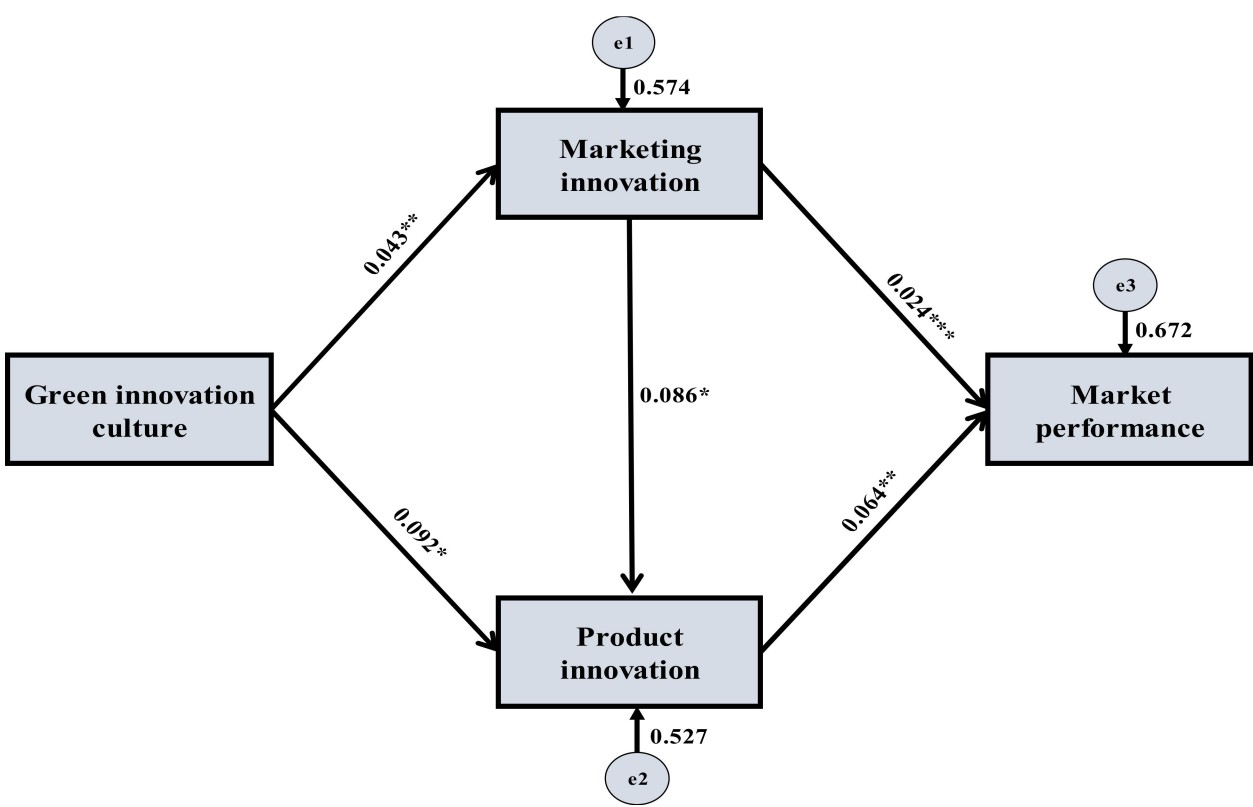

**Figure 2.** Schematic presentation of Structural equation modeling. *** *p* < 0.01, ** *p* < 0.05, * *p* < 0.1.

## 5. Conclusions and Policy Implications

This study examines how green innovation influences the market performance of small-and medium-sized enterprises (SMEs) in China. The study results show that innovation frameworks help SMEs succeed in the market. The structural equation modeling approach is used for data analysis on a sample of 453 respondents. The empirical results show that green innovation significantly influences marketing innovation and product innovation. Marketing innovation positively influences product innovation and market performance, whereas product innovation positively influences market performance.

### 5.1. Theoretical Implications

According to Terziovski's model, the success of an SME depends upon its structure, culture, values, and capabilities. This paper discusses the success of SMEs in Chin in light of GIC, MRK, and PRD. Thus, this study's results are intended to add to the existing work on organizational practices, SMEs, and innovation in China. It substantially adds to the studies on the MRP of SMEs by examining in detail their innovative structures. The theoretical model details the connection between innovation culture, PRD, market innovation, and MRP of SMEs. The results show that a culture of innovation leads to marketing and PRD (H1 and H2). Furthermore, some valid conclusions from these findings include that innovation culture is a precondition for administrative, advertising, and organizational success in competitive markets. Though the prior studies [35,36,41] have demonstrated the value of an organization's innovation culture, several concerns about the links between innovation culture and SMEs' innovative marketing tactics remain unanswered.

The innovation culture of SMEs in China positively impacted the marketing aspect of things and PRD and performance. When an organization's innovation culture is rich and diverse, it can not only encourage the development of new ideas and products but can also come up with interesting marketing strategies that catch the eye of the customer. Moreover, a culture of innovation often aids in product development processes too. According to the research, the MRK tactic has an important and favorable association with both PRD

and MRP (H3 and H4). According to previous studies, MRK substantially influences ENS, corporate performance, and SME success.

On the other hand, this study builds upon prior research by testing MRK in an integrated model, emphasizing SME and ENS. When new items' development and marketing operations are well-executed, they are successful. When a product is first introduced to the market, potential customers have limited knowledge of it. As a result, businesses will require the latest products to present and sponsor them, resulting in MRK.

Numerous studies indicate that PRD is essential for developing new products, process efficiencies, and increasing market share on a long-term basis. According to the study, PRD has a strong and significant linkage with MRP (H5). In addition, one-of-a-kind added items have the impact of improving operation.

The paradigm findings reveal that innovation culture and MRK are directly linked to PRD in SMEs. The study's findings provide academics with a useful viewpoint, stating that GIC encourages SMEs to make their products distinct from their competitors. As a result, this research adds to the existing work on innovation by better understanding the link between innovation and SMEs' market success. Specifically, it examines the influence of marketing and PRD on MRP to further our knowledge of the association between innovation and ENS.

*5.2. Managerial Implications*

This study highlights certain repercussions for administrators of SMEs for the importance of PRD, MRK strategies, and expanding a company's base in the market. For beginners, SMEs should work on their MRK rigorously to achieve a viable edge by cultivating a culture of innovation in the organization. In the case of developing new and innovative products, SME management must develop new goods and achieve outstanding ENS.

The study also recommends that SMEs try to stabilize their assets for an innovative learning culture, marketing, and innovation processes to promote ENS. These findings aid managers in achieving greater market results. Finally, SMEs should invest in promotion strategies and implement new marketing programs inside their organizations to boost their PRD capabilities. In addition, SMEs should be responsive to this type of innovation because of their information technology environment and brand management operations, since enhancing this competency to encourage innovation would boost ENS. As can be observed, the strategy described in the study allows managers to take a novel perspective on how SMEs integrate marketing and creative skills to achieve successful ENS. Marketing and product development may benefit from incorporating an innovation culture into the organization's structure. Consequently, managers may influence employee behavior, conduct, and the incorporation of new ideas to promote ENS.

## 6. Limitations and Potential Research

This research has several limitations, most related to sample size and design. It first examined how SME CEOs felt about MRP and product and service innovation. The results may be further developed using objective performance measurements in future studies. Second, the survey discussed above was sent to emails that the CoC and CoI already have for their members in their databases. Nevertheless, there is no reason to believe this influenced the firms' choice to participate in the study. It first examined how SME CEOs felt about MRP and product and service innovation. When examining organizational culture, future scholars should look at some more determinants, such as organizational innovation.

The role of managers in fostering marketing and PRD via creating an innovation culture is a potential subject for study. On the other hand, the process of building innovative dimensions in SMEs is worth investigating. Managers' actions will encourage many sorts of innovation within the company. Managers must communicate to employees the critical aspects of conducting market-based, client-focused innovation studies. As a result, future research may focus on marketing and product enhancements that influence and encourage

ENS. The example is a good start, but it is not thorough. The model's extension will yield additional information. Future research prospects include broadening the scope of marketing. In case of researching the significance of MRP in small enterprises, innovation and PRD on ENS, scholars in the coming time must look into some more aspects, such as company innovation and company culture, by including other factors, for example, innovation of process and knowledge patterns, both of which can assist both academics and administrators.

**Author Contributions:** Conceptualization, S.Z.; Funding support, X.Y.; Writing — review & editing, X.Y.; Funding support, W.G.; Data curation, W.G.; Formal analysis, Y.Y.; Methodology, J.L.; Writing — original draft, B.L. All authors have read and agreed to the published version of the manuscript.

**Funding:** This research was supported by the following funds: China Postdoctoral Science Foundation: A study on the mechanism of physician engagement behaviour in online medical communities from the perspective of network effects (No. 2022M710038). Guangxi Science and Technology Base and Talent Special Project: Research on the incentive mechanism of user information sharing in live e-commerce-based on social capital perspective (No., 2020AC19034). 2021 Guangxi 14th Five-Year Education Science Planning Key Special Project: Research on the influence of learning communities on users' online learning behavior in the information technology environment (No., 2021A033). 2021 Guangxi 14th Five-Year Education Science Planning Key Special Project: Research on the influence of short video sharing on Chinese cultural identity of international students in China-taking Jieyin as an example (No., 2021ZJY1607). 2022 Innovation Project of Guangxi Graduate Education: Research on Cultivating Innovation and Practical Ability of Postgraduates in Local Universities in Guangxi. (No., JGY2022122). Guangxi undergraduate teaching reform project in 2022: research on the construction of thinking and government in marketing courses under the online and offline mixed teaching mode. (No., 2022JGB180). Teaching reform project of Guilin University of Electronic Science and Technology: research on the construction of the ideology and politics of the course of Brand Management. (No., JGB202114). Doctoral research initiation project of Guilin University of Electronic Science and Technology: "Research on the incentive mechanism of knowledge sharing in online medical communities" (No., US20001Y).

**Institutional Review Board Statement:** The study was conducted in accordance with the Declaration of Helsinki and approved by the Institutional review board of Guilin University of Electronic Technology (protocol code 893-2, 12-03-2022).

**Informed Consent Statement:** Informed consent was obtained from all subjects involved in the study.

**Data Availability Statement:** Not applicable.

**Conflicts of Interest:** There are no conflict of interest.

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
