# Peer review of "Assessing the Influence of Green Innovation on the Market Performance of Small- and Medium-Sized Enterprises"

_sustainability, doi:10.3390/su142012977_

Round 1
Reviewer 1 Report
The paper's topic and conducted research are very important and justified to be presented in a high-quality Journal. The subject is very important for the literature. However, some issues need to be addressed carefully. My decision is – a minor revision, with some amendments. Please see my comments and suggestions below.
Point 1. The title could be changed to Assessing the Influence of Green Innovation on Market Performance of Small and Medium-Sized Enterprises.
Point 2. The authors should point out the research gap for this study in the Introduction.
Point 3. The authors should have more literature to support the proposed hypotheses, such as H1b and H2b. The following papers can be good examples to help you improve your paper:
- Qing, L.; Chun, D.; Dagestani, A.A.; Li, P. Does Proactive Green Technology Innovation Improve Financial Performance? Evidence from Listed Companies with Semiconductor Concepts Stock in China. Sustainability 2022, 14, 4600. https://doi.org/10.3390/su14084600
- Qing, L.; Chun, D.; Ock, Y.-S.; Dagestani, A.A.; Ma, X. What Myths about Green Technology Innovation and Financial Performance's Relationship? A Bibliometric Analysis Review. Economies 2022, 10, 92. https://doi.org/10.3390/economies10040092
- Ma, X., Ock, Y. S., Wu, F., & Zhang, Z. The Effect of Internal Control on Green Innovation: Corporate Environmental Investment as a Mediator. Sustainability 2022, 14(3), 1755. https://doi.org/10.3390/su14031755
Point 4. And discussion? Discussion should be the evaluation of the obtained results (author's original thoughts) in the light of the previous research (could include the items as the explanation of the hypothesis).
Point 5. Rewriting conclusions according to the purpose of the paper. Please, transparently explain the research results in the 1st paragraph of the conclusion.
Point 6. In 5.1 Theoretical Implications, the authors should describe which research gaps the results of this study fill in the previous literature, respectively.
Good luck for your work!

Author Response
The paper's topic and conducted research are very important and justified to be presented in a high-quality Journal. The subject is very important for the literature. However, some issues need to be addressed carefully. My decision is – a minor revision, with some amendments. Please see my comments and suggestions below.
Comment: The title could be changed to Assessing the Influence of Green Innovation on Market Performance of Small and Medium-Sized Enterprises.
Authors' response: Dear reviewer, Respected reviewer, thank you very much for the positive evaluation of this study. We are very honored and pleased that you have taken a particular interest in this manuscript. We appreciate your dedication and time in reviewing this submission. Aiming to improve the manuscript, we have considered your suggestion and chnages the topic as follows:
“Assessing the Influence of Green Innovation on Market Performance of Small and Medium-Sized Enterprises”
Comment: The authors should point out the research gap for this study in the Introduction.
Authors' response: Thank you for your keen observation and constructive feedback, which greatly helped us improve the revised draft. We have carefully looked into this concern and clearly mention research gaps in the introduction section. Please refer to the revised introduction section in this regard.
Comment: The authors should have more literature to support the proposed hypotheses, such as H1b and H2b. The following papers can be good examples to help you improve your paper:
- Qing, L.; Chun, D.; Dagestani, A.A.; Li, P. Does Proactive Green Technology Innovation Improve Financial Performance? Evidence from Listed Companies with Semiconductor Concepts Stock in China. Sustainability 2022, 14, 4600. https://doi.org/10.3390/su14084600
- Qing, L.; Chun, D.; Ock, Y.-S.; Dagestani, A.A.; Ma, X. What Myths about Green Technology Innovation and Financial Performance's Relationship? A Bibliometric Analysis Review. Economies 2022, 10, 92. https://doi.org/10.3390/economies10040092
- Ma, X., Ock, Y. S., Wu, F., & Zhang, Z. The Effect of Internal Control on Green Innovation: Corporate Environmental Investment as a Mediator. Sustainability 2022, 14(3), 1755. https://doi.org/10.3390/su14031755
Authors' response: Thank you again for your encouragement and observation. We appreciate your suggestion to further strengthen the contents and quality of this study. The suggested studies helped us a lot in revising the manuscript. Following your suggestion, we have revised the literature review section and cited the above-mentioned studies at suggested places. Please refer to the revised manuscript for more details.
Comment: Discussion should be the evaluation of the obtained results (author's original thoughts) in the light of the previous research (could include the items as the explanation of the hypothesis).
Authors' response: Thank you very much for the valuable suggestion. In light of your valuable feedback, we have significantly revised the discussion section and compared it with other research works to further improved the significance of the current study.
Comment: Rewriting conclusions according to the purpose of the paper. Please, transparently explain the research results in the 1st paragraph of the conclusion.
Authors' response: We appreciate your keen observation. As suggested, we have revised the conclusion section and research results have been mentioned in the first paragraph of the conclusion. Please refer to the revised conclusion section for more details.
Comment: In 5.1 Theoretical Implications, the authors should describe which research gaps the results of this study fill in the previous literature, respectively.
Good luck for your work!
Authors' response: We appreciate the concern of the worthy reviewer. In the revised version of the manuscript, this issue has been resolved by clearly mentioning how the results of current study fill the research gaps in the previous literature.
After responding to the expert reviewer's valuable comments and incorporating useful suggestions, the authors are confident that this study in the current revised version has been improved to the best possible point compared to the previous version, which will satisfy the concerns raised by the worthy reviewer. Once again, thank you very much for your professional review of our manuscript. We are looking forward to hearing from you soon.
**************************************************************
Reviewer 2 Report
The proposed paper is very interesting.
*Please justify in the paper to what extent the developed paper can be relevant to the international scientific field.
*In the conclusion, please indicate to what extent the paper is innovative and what is its scientific contribution in the field of Green Innovation
Author Response
Comment: The proposed paper is very interesting. Please justify in the paper to what extent the developed paper can be relevant to the international scientific field.
Authors' response: Dear reviewer, thank you very much for the positive evaluation of our manuscript. We greatly appreciate the time and energy you dedicated to reviewing this submission. Aiming to improve the manuscript, we have considered your feedback and made all of the suggested revisions. In this regard, the contributions and novelty of the current study has been clearly highlighted in the revised manuscript. Following revisions have been made in this regard:
In light of the preceding debate, this study aims to look at the link between Chinese innovation and MRP in China. The research will also analyze the resource-based perspective (RBV) based on Terziovski's work. Unlike prior research, this focuses on MRP as a factor in a company's success (Li and Choi, 2020). This research tries to provide clear knowledge of innovation paradigms and how they generate ENS, particularly in China. The first goal is to emphasize the relevance of marketing innovation (MRK) and green in-novation culture (GIC) in SMEs regarding product innovation (PRD). It's important to re-member that innovation is necessary at all phases of competition and that it generates riches in the commercial sector (Rubio-Andrés, 2022). According to various studies, small businesses spend more money on product development than process development. As a result, this research is entirely concerned with the impact of PRD on MRP (Gherghina, 2020).
To grow by generating new goods and services, businesses must include an innovation culture in their operations. Creativity, empowerment, and a transformation in the company culture are all catalysts for innovation. Empirical research shows that a culture of innovation must be established, maintained, and encouraged if businesses are successful and generate new goods (Martins & Calado, 2018). Although the literature focuses on MRK and GIC, the importance of an innovative culture and the influence of MRK on PRD have not been properly investigated in the previous study (Sprong, 2021).
The study's second purpose is to determine the importance of MRK methods and PRD in attaining greater ENS. The basic assumption of this study is that MRK is required while attempting to improve MRP (Hussain, 2020). The primary contributors to MRP are marketing and PRD strategies. Competition has become a necessary component of market survival, whereas innovation activities generate higher value and advantages, such as helping a firm stand out from its competitors (Haanaes, 2020). In complicated circum-stances, SMEs can effectively employ market innovation to sell distinctive products and services. According to the literature on innovation, PRD impacts performance (Dawar & Frost, 2014).
The study's focus is thus on the influence of creative activities, such as innovation, on ENS. This research contributes to the current body of knowledge by revealing how SMEs may succeed in PRD by cultivating a distinct innovation culture and MRK (Her-mundsdottir & Aspelund, 2021). It also claims that SMEs outperform large corporations in the marketplace owing to the creation of innovative marketing tactics and goods (Tuomi-nen, 2022). The study will take a theoretical approach that an RBV is essential for ex-plaining and resolving the model's future issues. More specifically, the current study will address the following research questions: (i), how do market and PRD activities affect SMEs' ENS? (ii) What impact does SMEs' entrepreneurial culture have on marketing and product development? (iii) How do marketing and PRD combine to impact consumer behavior?
Comment: In the conclusion, please indicate to what extent the paper is innovative and what is its scientific contribution in the field of Green Innovation.
Authors' response: We appreciate your suggestion to further improve the contents and quality of the current study. Following your suggestion, we have revised the conclusion section by clearly mentioning the innovativeness and scientific contribution of the current study related to green innovation. Following revisions have been made in this regard:
This study examines how green innovation influence the market performance of small and medium-sized enterprises (SMEs) in China. The study results show that innovation frameworks help SMEs succeed in the market. The structural equation modeling approach is used for data analysis on a sample of 453 respondents. Empirical results show that green innovation significantly influence marketing innovation and product in-novation. Marketing innovation positively influence product innovation and market performance, whereas product innovation positively influences market performance.
According to Terziovski's model, the success of an SME depends upon its structure, culture, values, and capabilities. In this paper, the success of SMEs in Chin has been dis-cussed in light of GIC, MRK, and PRD. Thus, this study's results are intended to add to the existing work on organizational practices, SMEs, and innovation in China. It adds to a great extent to the studies on the MRP of SMEs by examining in detail their innovative structures. The theoretical model details the connection between innovation culture, PRD, market innovation, and MRP of SMEs. The results show that a culture of innovation leads to marketing and PRD (H1a and H1b). Furthermore, some valid conclusions from these findings include that in competitive markets, an innovation culture is a precondition for administrative, advertising, and organizational success. Though the prior studies of (Ma et al., 2022; Qing et al., 2022 a,b) have demonstrated the value of an organization's innovation culture, several concerns about the links between innovation culture and SMEs' innovative marketing tactics remain unanswered.
The innovation culture of SMEs in China had a positive impact on the marketing aspect of things and PRD and performance. When an organization's innovation culture is rich and diverse, it can not only encourage the development of new ideas and products but can also come up with interesting marketing strategies that catch the eye of the customer. Moreover, a culture of innovation often aids in product development processes too. According to the research, the MRK tactic has an important and favorable association with both PRD and MRP (H2a and H2b). According to previous studies, MRK has a substantial influence on ENS, corporate performance, and SME success.
On the other hand, this study builds on prior research by putting MRK to the test in an integrated model, emphasizing SME and ENS. When new items' development and marketing operations are well-executed, they are successful. When a product is first introduced to the market, potential customers have limited knowledge of it. As a result, businesses will require the latest products to present and sponsor them, resulting in MRK.
Numerous studies indicate that PRD is essential for the development of new products, process efficiencies, and the increase of market share on long term-basis. According to the study, PRD has a strong and significant linkage with MRP (H3). In addition, one-of-a-kind added items have the impact of improving operation.
The findings of the paradigm reveal that in SMEs, innovation culture and MRK are directly linked to PRD. The study's findings provide academics with a useful viewpoint, stating that GIC encourages SMEs to make their products distinct from their competitors. As a result, this research adds to the existing work on innovation by better understanding the link between innovation and SMEs' market success. Specifically, it examines the in-fluence of marketing and PRD on MRP to further our knowledge of the association be-tween innovation and ENS.
Th study further highlights certain repercussions for administrators of SMEs for the importance of PRD, MRK strategies, and expanding a company's base in the market. For beginners, SMEs should work on their MRK rigorously to get a viable edge by cultivating a culture of innovation in the organization. In the case of developing new and innovative products, SME management must develop new goods and achieve outstanding ENS.
The study's findings also recommend that SMEs try to stabilize their assets for an innovative learning culture, marketing, and innovation processes to promote ENS. These findings aid managers in achieving greater market results. Finally, SMEs should invest in promotion strategies and implement new marketing programs inside their organizations to boost their PRD capabilities. In addition, SMEs should be responsive to this type of in-novation because of their information technology environment and brand management operations, since enhancing this competency to encourage innovation would boost ENS. As can be observed, the strategy described in the study allows managers to take a novel perspective on how SMEs integrate marketing and creative skills to achieve successful ENS. Marketing and product development may benefit from incorporating an innovation culture into the organization's structure. Consequently, managers may influence employ-ee behavior, conduct, and the incorporation of new ideas to promote ENS.
After incorporating all of the worthy reviewers' suggestions and concerns, the authors are confident that the paper in its current revised version is improved to the best possible point compared to the previous version, which will satisfy the concerns raised by the respected reviewer. Once again, thank you very much for your professional review of our manuscript. We are looking forward to hearing from you soon.
************************************************************
Reviewer 3 Report
The manuscript is subject to the following revisions:
Abstract: The abstract should be written in the following context: Background, objective(s), methods, results, conclusions, policy recommendations.
How you can compare your results with other studies of same geographical regions. It is pivotal to tie your results with other relevant literature.
Discussion: The study discussion is very generic and do not stem from study findings. It is suggested to rearrange discussion based on study findings.
Conclusions: Conclusions are weak and miss several important dimensions. To strengthen the contents and quality of the study, conclusions must be revised for more clarity and for the ease of normal readers.
Policy recommendations: Specific policy recommendations should be put forward according to the target sample. General policies are of no use in scholarly articles.
Study limitations should be provided along with future research directions for prospective scholars interested in the similar works.
The authors have used several old references to support their arguments. We are in 2022 and you are using such old references. In order to nurture the importance of study, references should be updated using recent and relevant studies.
There is an intermingle of capital and small letters. Please avoid this practice in scientific writing.
Finally, the manuscript can be benefited if the authors thoroughly proofread it in terms of English language mistakes and syntax structure.
Author Response
The manuscript is subject to the following revisions:
Comment: Abstract: The abstract should be written in the following context: Background, objective(s), methods, results, conclusions, policy recommendations.
Authors' response: Dear reviewer, thank you very much for the positive evaluation of our manuscript. As suggested, we have revised the abstract.
Comment: How you can compare your results with other studies of same geographical regions. It is pivotal to tie your results with other relevant literature.
Authors' response: Thank you very much for the concern. As demanded, we have tied our results with previous literature.
Comment: Discussion: The study discussion is very generic and do not stem from study findings. It is suggested to rearrange discussion based on study findings.
Authors' response: Thank you very much for the concern. As demanded, we have rearranged the discussion part.
Comment: Conclusions: Conclusions are very weak and miss several important dimensions. To strengthen the contents and quality of the study, conclusions must be revised for more clarity and for the ease of normal readers.
Authors' response: Thank you very much for the concern. We have revised the conclusions with more clarity.
Comment: Policy recommendations: Specific policy recommendations should be put forward according to the target sample. General policies are of no use in scholarly articles.
Author’s response: Thank you very much for the valuable suggestion. Following your suggestion, we have revised the policy recommendations.
Comment: Study limitations should be provided along with future research directions for prospective scholars interested in the similar works.
Authors' response: We appreciate your keen observation. We have provided the study limitations along with future recommendations.
Comment: The authors have used several old references to support their arguments. We are in 2022 and you are using such old references. In order to nurture the importance of study, references should be updated using recent and relevant studies.
Authors' response: We appreciate your keen observation. We have provided the most recent citations in the revised manuscript.
Comment: There is intermingle of capital and small letters. Please avoid this practice in scientific writing.
Authors' response: We appreciate your keen observation. We have avoided the intermingle of capital and small letters in the revised manuscript.
Comment: Finally, the manuscript can be benefited if the authors thoroughly proofread it in terms of English language mistakes and syntax structure.
Authors' response: We appreciate the worthy reviewer for highlighting this concern. In the revised manuscript, we have proofread all the English language mistakes and syntax structure.
*****************************